# Revealing New Landscape of Turbot (*Scophthalmus maximus*) Spleen Infected with *Aeromonas salmonicida* through Immune Related circRNA-miRNA-mRNA Axis

**DOI:** 10.3390/biology10070626

**Published:** 2021-07-06

**Authors:** Ting Xue, Yiping Liu, Min Cao, Mengyu Tian, Lu Zhang, Beibei Wang, Xiaoli Liu, Chao Li

**Affiliations:** School of Marine Science and Engineering, Qingdao Agricultural University, Qingdao 266109, China; xue2011ting124@126.com (T.X.); yipingliu414@163.com (Y.L.); caominjiayou@163.com (M.C.); mytianmengyu@126.com (M.T.); zhanglu19950918@163.com (L.Z.); w15288966538@163.com (B.W.); LXL17854231102@163.com (X.L.)

**Keywords:** turbot, spleen, *Aeromonas salmonicida*, triple regulatory networks, immune

## Abstract

**Simple Summary:**

In this study, the expression of circRNAs, miRNAs, and mRNA in the immune organs spleen of turbot (*Scophthalmus maximus*) infected with *Aeromonas salmonicida* was analyzed by high-throughput sequencing, and circRNA-miRNA-mRNA network was constructed, so as to explore the function of non-coding RNA in the immune system of teleost. A total of 119, 140, and 510 differential expressed circRNAs, miRNAs, and mRNAs were identified in the infected groups compared with the uninfected group. The qRT-PCR verified the reliability and accuracy of the Illumina sequencing data. Fifteen triple networks of circRNA-miRNA-mRNA were presented in the form of “up (circRNA)-down (miRNA)-up (mRNA)” or “down-up-down”. Immune-related genes were also found in these networks. These results indicate that circRNAs and miRNAs may regulate the expression of immune-related genes through the circRNA-miRNA-mRNA regulatory network and thus participate in the immune response of turbot spleen after pathogen infection.

**Abstract:**

Increasing evidence suggests that non-coding RNAs (ncRNA) play an important role in a variety of biological life processes by regulating gene expression at the transcriptional and post-transcriptional levels. Turbot (*Scophthalmus maximus*) has been threatened by various pathogens. In this study, the expression of circular RNAs (circRNAs), microRNAs (miRNAs), and mRNA in the immune organs spleen of turbot infected with *Aeromonas salmonicida* was analyzed by high-throughput sequencing, and a circRNA-miRNA-mRNA network was constructed, so as to explore the function of non-coding RNA in the immune system of teleost. Illumina sequencing was performed on the uninfected group and infected group. A total of 119 differential expressed circRNAs (DE-circRNAs), 140 DE-miRNAs, and 510 DE-mRNAs were identified in the four infected groups compared with the uninfected group. Most DE-mRNAs and the target genes of DE-ncRNAs were involved in immune-related pathways. The quantitative real-time PCR (qRT-PCR) results verified the reliability and accuracy of the high-throughput sequencing data. Ninety-six differentially expressed circRNA-miRNA-mRNA regulatory networks were finally constructed. Among them, 15 circRNA-miRNA-mRNA were presented in the form of “up (circRNA)-down (miRNA)-up (mRNA)” or “down-up-down”. Immune-related genes *gap junction CX*32.2, *cell adhesion molecule* 3, and *CC chemokine* were also found in these networks. These results indicate that ncRNA may regulate the expression of immune-related genes through the circRNA-miRNA-mRNA regulatory network and thus participate in the immune response of turbot spleen after pathogen infection.

## 1. Introduction

Turbot (*Scophthalmus maximus*, *S. maximus*) is a kind of cold-water benthic fish that originated from Europe, which is an important economic species in many countries. However, in recent years, the aquaculture of turbot was frequently interfered with by bacterial diseases, especially *Aeromonas salmonicida* (*A. salmonicida*), *Edwardsiella tarda* (*E. tarda*) and *Vibrio anguillarum* (*V. anguillarum*), which seriously restricted the development of turbot aquaculture [1]. Therefore, it is very urgent to study the immune system and disease mechanism of turbot.

Gene regulatory network is a complex dynamic network system, which is the overall performance of the interaction between various factors. In recent years, with the development of next-generation sequencing technology and the integration analysis and application of multi-omics data, a variety of non-coding RNAs (ncRNAs), such as long non-coding RNAs (lncRNAs), microRNAs (miRNAs), and circular RNAs (circRNAs), have been confirmed to be involved in the complex biological process of fish resistance to bacterial immunity.

CircRNAs are a class of single-stranded closed RNA with 3′ and 5′ ends linked to each other [2]. The circRNA was first discovered in 1979 using electron microscopy in eukaryotic cells [3]. CircRNAs were initially considered to be the abnormally spliced transcripts [4,5]. However, thousands of circRNAs have been found in various organisms [6,7,8,9]. With abundant miRNA binding sites, circRNAs can act as miRNA sponges by binding and adsorbing with miRNA [10]. MicroRNA is a short (~22 nt), endogenous, evolutionarily conserved non-coding RNA that can down-regulate the expression of a target gene by binding to the 3′ UTR of mature mRNA [11]. Therefore, the adsorption of circRNAs to miRNAs prevents the binding of miRNA to its target mRNAs, thus indirectly regulating the expression of miRNA target genes [12].

Recent studies showed that circRNAs are involved in a variety of biological processes, such as cell regulation [13], signaling pathway regulation [14], tissue development [15], and so on. Nevertheless, other numerous researches revealed that circRNAs also plays an important role in the occurrence and treatment of diseases such as tumors, tissue injury, and depression [16,17,18]. Furthermore, some of these studies revealed that circRNAs are also involved in innate immune responses [19,20]. CircRNA (102911) is involved in the proliferation and apoptosis of CD3(+)/CD4(+) T lymphocytes through adsorbing miR-129-5p [21]. A high level of circ_0020710 could upregulate the CXCL12 expression via sponging miR-370-3p and promotes melanoma growth in vivo [22]. Circ-RelL1 could regulate inflammatory response in ox-LDL-induced endothelial cells through miR-6873-3p/MyD88/NF-κB axis [23].

The identification of circRNAs in fish has also gradually increased, including those participating in the process of teleost resistance to pathogen infection [24,25,26]. In grass carp, 41 differential expressed circRNAs (DE-circRNAs) were identified after GCRV (Grass Carp Reovirus) infection, and these circRNAs may be related to the hemorrhagic symptoms [24]. circPIKfyve activated the NF-κB/IRF3 pathway by sponging miR-21-3p, thereby enhancing the innate antiviral response of teleost [25]. As a competitive endogenous RNA (ceRNA) of miR-21-3p, circrasgF1b alleviated the inhibitory effect of miR-21-3p on its target, *MITA*, and thus enhanced the innate anti- SCRV (Sininiperca Chuatsi Rhabdovirus) response [26].

In addition to the functions of circRNAs through translation and pseudogene derivation influencing the splicing of its linear cognates, regulating transcription and splicing, circRNAs mainly play their role through binding with miRNA [27]. CircRNAs affect the expression of mRNAs by combining with miRNAs, forming circRNA-miRNA-mRNA regulatory network, and then affecting downstream genes or pathways, thus playing a role in a variety of biological processes, such as autoimmune disease [28], stress response [29], cancer [30], and cardiovascular disease [31]. Emerging studies have shown that circRNA-miRNA-mRNA regulatory networks have also been identified in pathogen-infected teleost [9,32,33].

In this study, we examined the expression of circRNA, miRNA, and mRNA in the spleen of turbot infected with *A. salmonicida* at different time points by high-throughput sequencing. The differentially expressed circRNAs, miRNAs, and mRNAs were screened and the regulatory networks of circRNA-miRNA, miRNA-mRNA, and circRNA-miRNA-mRNA were constructed. Our aim is not only to broaden our understanding of turbot immune regulatory network, but also to provide new ideas to enhance turbot resistance to pathogenic bacteria invasion.

## 2. Materials and Methods

### 2.1. Turbot Treatment and Sample Collection

Healthy turbot (100 ± 10 g) were purchased from Huanghai Aquaculture Company (Haiyang, Shandong, China). The experimental protocols were approved by the Committee on the Ethics of Animal Experiments of Qingdao Agricultural University IACUC (Institutional Animal Care and Use Committee). The experiment was carried out after 2 weeks of temporary breeding in the laboratory. The feeding was stopped 3 days before the experiment. According to Coscelli et al., *A. salmonicida* was injected into turbot by intraperitoneal injection [34]. The *A. salmonicida* was first reactivated from the infected turbot and then cultured by LB culture medium. The bacteria were resuspended in PBS at a concentration of 1 × 10^7^ CFU and inoculated intraperitoneally at 0.1 mL per tail. The turbot spleen tissue samples from 15 fish injected with bacteria for 6 h (ASS6), 12 h (ASS12), 24 h (ASS24), and 96 h (ASS96) (five fish per pool) were sampled. All fish were anesthetized with 0.01% ethyl-m-aminobenzoate until breathing ceased before the samples were collected. The PBS-injected turbot was taken as control (ASS0). The extracted tissue samples were frozen in liquid nitrogen and subsequently analyzed by high-throughput sequencing.

### 2.2. Total RNA Isolation, Library Construction, and Sequencing

Total RNA was extracted from the collected samples with TRIzol reagent (Invitrogen, CA, USA) according to the manufacturer’s manual. The purity, concentration, and quality of the total RNA were measured using NanoPhotometer^®^ spectrophotometer (IMPLEN, CA, USA) and Qubit^®^ RNA Assay Kit in Qubit^®^ 2.0 Flurometer (Life Technologies, CA, USA). The integrity of RNA was accurately detected by Agilent 2100 bioanalyzer system (Agilent Technologies, CA, USA).

The preparation steps of mRNA and circRNA libraries followed manufacturer’s recommendations of NEBNext^®^ UltraTM RNA Library Prep Kit for Illumina^®^ (NEB, USA). First, after the RNA was tested and qualified, 3 μg RNA was prepared for each sample and ribosomal RNA (rRNA) was removed from the total RNA using the Epicenter Ribo-Zero rRNA Removal Kit (Illumina, WI, USA). mRNA was purified from total RNA using poly-T oligo-attached magnetic beads. Different from mRNA library, the construction of circRNA library required the addition of 40U RNase R to the rRNA removal system and incubation at 37 °C for 3 h to remove the linear RNA. Next, the treated RNA samples were fragmented to 250–300 bp.

The first strand cDNA was synthesized using random hexamer primer and M-MuLV Reverse Transcriptase (RNase H^−^). The DNA Polymerase I was then used for the synthesis of second strand cDNA. The library fragments were purified and selected using AMPure XP system (Beckman Coulter, CA, USA). The size-selected, adaptor-ligated cDNA was degraded by USER enzyme (NEB, MA, USA). Then PCR was performed to obtain the chain specific cDNA library and the quality of the purified library was assessed on the Agilent Bioanalyzer 2100 system.

The NEBNext^®^ Multiplex Small RNA Library Prep Set for Illumina^®^ (NEB, MA, USA) was used to construct the miRNA library. In brief, approximately 3 μg of total RNA was used as the initial sample, based on the special structure of sRNA with hydroxyl group at the 3′ end and phosphate group at the 5′ end; sRNA was specificity ligated with adaptors at both ends, and then the cDNA was synthesized using M-MuLV Reverse Transcriptase (RNase H^−^). Sequencing adaptors were ligated to double-stranded cDNA using PCR and the final miRNA library contains 140–160 bp fragments that were separated and recovered from an 8% polyacrylamide gel. The library quality was detected on the Agilent Bioanalyzer 2100 system using DNA High Sensitivity Chips. At last, the mRNA, circRNA, and sRNA library were clustered and then sequenced on an Illumina Hiseq 2500/2000 platform.

### 2.3. Data Analysis

#### 2.3.1. Quality Control

Clean data was generated by removing reads with an adapter, reads with ploy-N, and low quality reads from raw data. Simultaneously, Q20, Q30, GC-content, and sequence duplication level of the clean data were calculated. All the downstream analyses were based on the high-quality clean data. Then, the clean data was aligned to the assembled genome of *S. maximus* using STAR (v2.5.1b) or Bowtie [35].

#### 2.3.2. RNA Identification and Annotation

In consideration of the high false positives in circRNA identification, two software (find_circ and CIRI2) [36,37] were used for the identification of circRNA and the results from these two software were intersected.

The mapped small RNAs were first identified in the miRBase 20.0 database to find known miRNAs. Repeatmasker and RFAM software were used to remove the possible rRNA, tRNA, small nuclear RNA (snRNA), and small nucleolar RNA (snoRNA). Novel miRNA was analyzed using miREvo and mirdeep2 software [38,39], which depended on the hairpin structure of miRNA precursors. A total rRNA ratio of less than 40% was considered as a sample quality indicator. The base bias of seed region was calculated using Perl scripts. miFam.dat (http://www.mirbase.org/ftp.shtml, accessed on 3 September 2019) and Rfam (http://rfam.sanger.ac.uk/search/, accessed on 5 September 2019) database were used to look for families of known miRNA and novel miRNA precursors, respectively. miRanda [40] was used to predict the miRNA binding sites to target genes.

#### 2.3.3. Quantification of Gene Expression Levels

The expression levels of mRNA, circRNA, and miRNA were estimated using TPM (number of transcripts per million clean tags) [41]. DESeq2 R package (1.10.1) [42] was used for the different expression analysis of mRNA, circRNA, and miRNA under the condition of threshold *p*-value < 0.05.

#### 2.3.4. GO and KEGG Analyses

GOseq R package [43] and KOBAS (http://www.genome.jp/kegg/, accessed on 15 September 2019) were used to perform Gene Ontology (GO) and Kyoto Encyclopedia of Genes and Genomes (KEGG) enrichment analyses to further explore the function of differential expressed mRNA (DE-mRNA), DE-circRNA, or DE-miRNA in *S. maximus*.

#### 2.3.5. Co-Expression Network Analysis

A plurality of miRNA-mRNA and circRNA-miRNA interaction pairs were generated by target prediction of that miRNA to the circRNA and the mRNA. According to the miRNA that targeted circRNA and mRNA simultaneously, circRNA-miRNA-mRNA networks were constructed by Cytoscape 3.4.0 software [44]. In the established circRNA-miRNA-mRNA network, the expression trend of “up (circRNA)-down (miRNA)-up (mRNA)” or “down-up-down” was retained. The immune-related circRNA-miRNA-mRNA networks will be selected as candidates for future study.

#### 2.3.6. Validation the Expression of RNAs with Quantitative Real-Time PCR (qRT-PCR)

To confirm the reliability of sequencing data, four DE-circRNAs, four DE-miRNAs, and eight DE-mRNAs were randomly selected to verify their expressions by qRT-PCR. Oligo7 software was used for the specific primers design. The design of circRNA divergent primers spanned the circRNA backsplice junction. The forward primers of miRNA were designed according to the instructions of miRcute miRNA isolation kit (Tiangen Biotech, China). All the primers used in this study are listed in Appendix A. The total RNA extracted forms of *A. salmonicida*-infected and control *S. maximus* spleen were reverse-transcribed into cDNA by using the PrimeScript™ RT reagent Kit and 6 mers primers (Takara, Japan) for circRNAs and mRNAs detection. The total RNA extracted forms of *A. salmonicida*-infected and control *S. maximus* spleen were reverse-transcribed into cDNA by using the miRcute miRNA First-Stand cDFNA Synthesis Kit (Tiangen Biotech) for miRNA detection. qRT-PCR was performed on a CFX 96 real-time PCR system (Bio-Rad, Hercules, CA, USA) with SYBR^®^ Premix Ex TaqTM II (TliRNaseH Plus) (Takara, Japan).

The 18S rRNA was used as an internal control of circRNAs and mRNAs and U6 was used as an internal control of miRNA. Relative expression level was calculated using the 2^−ΔΔCt^ method [45]. All experiments were performed in triplicate and the data were shown as means ± SE. Correlation analysis of differential expression of RNAs in qRT-PCR and high-throughput sequencing was performed using Spearman’s nonparametric correlation (SPSS20.0).

## 3. Results

### 3.1. Statistical Analysis of circRNAs Data

In rRNA-depleted non-infected (ASS0) and infected libraries from different time points (6, 12, 24, and 96 h), average 87.30, 86.59, 96.44, 94.79, and 94.98 million clean reads were identified, respectively (Table 1). The results showed that the quality of the clean reads of all libraries was high, with Q20 > 95% and Q30 > 90%, and most (>85%) clean reads were mapped to the reference genome (Table 1).

CircRNAs were identified in all samples, and they were widely distributed on different chromosomes (Figure 1A and Appendix A). Of all the identified circRNAs, 78.56% were located in exons, 5.92% in introns, and 15.52% in intergene regions (Figure 1B). The size of most circRNAs ranged from 200 to 400 nt, and 82.27% of them had a predicted spliced length of <400 nt (Figure 1C). As shown in Figure 1D and Appendix A, a total of 28, 34, 42, and 47 DE-circRNAs were identified at 6 h, 12 h, 24 h, and 96 h relative to control, respectively. Of these, 59 circRNAs were up-regulated and 60 were down-regulated. Further, we performed a hierarchical cluster analysis and Venn diagram of all DE-circRNAs (Figure 1E,F). The results showed that the expression patterns of uninfected and *A. salmonicida*-infected samples were classified into different clusters (Figure 1E). Nevertheless, the types of DE-circRNA were changed with the increase of infection time. One circRNA (novel_circ_0005673) was shown differentially expressed at all four comparisons in the Venn diagram (Figure 1F).

Go and KEGG were performed to elucidate the biological function of circRNAs in *S. maximus* after *A. salmonicida* infection. The GO terms of all DE-circRNAs from four time points post-infection were classified into three types, biological processes, molecular functions, and cellular components (Figure 1G). The biological processes include cellular process, metabolic process, response to stimulus, signal transduction, cell communication, and cytokinesis. Protein binding, molecular function regulator, identical protein binding, and transmembrane receptor protein were the important functional molecules in *A. salmonicida* infection. Extracellular region, cell periphery, cell cortex, envelope, and outer membrane were functionally enriched in the cellular component. KEGG pathways analysis showed that the DE-circRNAs were involved in multiple biological processes, such as TGF-*β* signaling pathway, melanogenesis, tight junction, and adherens junction (Figure 1H).

### 3.2. Statistical Analysis of miRNAs Data

After removing the adaptor-contained and low-quality raw reads, approximately 44.15, 48.26, 40.76, 46.50, and 51.29 million clean reads were identified in 15 sRNA libraries (ASS0, ASS6, ASS12, ASS24, ASS96), respectively (Table 2). In order to analyze the distribution of small RNA on the genomes, filtered sRNA was compared with reference genomes. The results showed that 93.20%, 93.08%, 92.76%, 85.98%, and 94.72% small RNA reads of the five samples were successfully mapped onto the *S. maximus* genome (Table 2). Clean reads of each sample with a length range of 18~35 nt were filtered for subsequent analysis. From the statistical diagram of the length distribution of sRNA, the length of sRNA from the five groups varieties ranged from 21 to 23 nt (Figure 2A). rRNA, tRNA, snRNA, and snoRNA were annotated with Rfam and removed from sRNA.

The known miRNAs were analyzed by the miRBase database and novel miRNAs were predicted through the signature hairpin structure of miRNA precursors, respectively. A total of 910 miRNAs were identified, including 252 known miRNAs and 668 novel miRNAs (Table 3). The expression levels of known and novel miRNAs in each sample were counted and normalized by TPM. The overall expression pattern of miRNAs among the five groups was exhibited in Figure 2B and was highly consistent. Among the 910 miRNAs, 140 miRNAs showed significant changes in expression compared with the control group. There were 44, 63, 39, and 96 miRNAs differentially expressed in ASS6, ASS12, ASS24, and ASS96 compared with ASS0, respectively (Figure 2C). Hierarchical cluster analysis revealed that the DE-miRNAs formed obvious clustering, and there were significant differences between the uninfected and *A. salmonicida*-infected group (Figure 2D). As can be seen from the Venn diagram, one part of the miRNAs was differentially expressed at different time points of infection, and the other parts of the miRNAs were differentially expressed only at certain time points after infection (Figure 2E). More miRNAs were induced at 96 h after infection (Figure 2E).

The function of these DE-miRNAs was predicted through GO and KEGG analysis of their target genes. The GO terms of all target genes were also classified into three types. Most of the target genes involved biological processes, such as response to stimulus, signal transduction, cell communication, response to stimulus, and metabolic process. The majority of genes were important components of nucleoplasm, organelle inner membrane, mitochondrial inner membrane, and microtubule organizing center. These target genes mainly had a protein binding function, receptor activity, tumor necrosis factor receptor binding, and MHC protein binding function (Figure 2F). KEGG analysis of differentially expressed miRNAs showed that DE-miRNAs were annotated to wnt signaling pathway, TGF-β signaling pathway, MAPK signaling pathway, and adherens junction (Figure 2G).

### 3.3. Statistical Analysis of mRNAs Data

Through comparison, splicing, and screening, the expression levels of all transcripts were quantified, and 243, 187, 126, and 232 differential expressed genes were identified in four infected groups (ASS6, ASS12, ASS24, ASS96) compared with control group, respectively (Figure 3A). With the increase of infection time, the types of differentially expressed genes also changed (Figure 3B). Among them, 510 genes were differentially expressed at two or more infected time points, and 20 genes were differentially expressed at all four infected time points (Figure 3C).

Go analysis was performed to analyze the biological function of DE-mRNAs. The results showed that as the components of the cell membrane, extracellular region, or extrinsic component of membrane, these DE-mRNAs were mainly involved in a variety of biological processes, such as biological regulation, response to stimulus, cell communication, signal transduction, and transport. In these biological processes, they exhibited important functions in transporter activity, transmembrane transporter activity, signal transducer activity, receptor activity, and so on (Figure 3D). KEGG analysis showed that DE-mRNAs were annotated to various metabolic pathways, including glycolysis/gluconeogenesis, purine metabolism, carbon metabolism, RNA degradation, and biosynthesis of amino acids (Figure 3E).

### 3.4. Construction of the circRNA-miRNA Regulatory Network

CircRNAs regulate the expression of mRNAs by acting as miRNA sponges to antagonize the inhibition of miRNA on their target genes. The close relation of miRNA to the occurrence of disease by regulating mRNA expression suggests that circRNA also plays an important regulatory role in the occurrence of disease. Therefore, circRNA-miRNA regulatory networks were constructed to analyze the role of circRNAs as miRNA sponges in the process of *A. salmonicida* infection. A total of 1051 circRNA-miRNA interactions including 116 DE-circRNAs and 290 miRNAs were identified in the four infected groups (Appendix A).

Analysis showed that a single circRNA could adsorb more than one miRNA, while a single miRNA could bind to more than one circRNA. For example, novel_circ_0002735 could adsorb 28 miRNAs, and dre-miR-1306 could bind to 18 circRNAs. These miRNAs targeted a variety of immune-related genes, such as dre-miR-15b-5p targeted *interleukin*-20, dre-miR-138-2-3p targeted *interleukin*-8, dre-miR-138-5p targeted *interferon regulatory factor*-4, dre-miR-133a-3p and dre-miR-133a-3p targeted *C-X-C motif chemokine* 14, and dre-miR-133b-3p targeted *tumor necrosis factor β* (Appendix A). This finding suggested that these circRNA-miRNA pairs act as important regulatory mediators during the infection of *A. salmonicida*.

### 3.5. Construction of the miRNA-mRNA Regulatory Network

MicroRNAs usually downregulate gene expression in a variety of manners through partially complementary to one or more mRNA molecules. The miRNA-mRNA regulation networks were constructed to analyze the main functions of miRNA-mRNA pairs in *S. maximus* infected with *A. salmonicida*. In four infected groups, a total of 9605 mRNAs that were identified could be targeted by 140 DE-miRNAs (Appendix A). The absolute majority of miRNAs could target more than one mRNA, while only 10 miRNAs specifically targeted a single mRNA, such as dre-miR-126a-3p, dre-miR-203a-3p, dre-miR-142a-5p, dre-miR-1, and novel_1136. Similarly, the same one mRNA could also be bound by multiple or single miRNAs. The results showed that the mRNAs targeted by these miRNAs contained a variety of immune-related genes, including tight junction protein, interferon regulatory factor, interleukin, complement component, tumor necrosis factor receptor, integrin, C-X-C motif chemokine, gap junction, NF-κB inhibitor, and so on. *C-X-C motif chemokine* 14 could be targeted by dre-miR-133a-3p, dre-miR-133b-3p, and dre-miR-210-3p; *gap junction α*-5 could be targeted by novel_513; *NF-κB inhibitor δ* could be targeted by dre-miR-107b; *complement C1q* could be targeted by novel_753 and novel_880; *interleukin*-17 *receptor* B, *tight junction protein ZO*-1, and *NF-κB inhibitor ζ* all could be targeted by novel_324. The above results suggested that miRNA-mRNA pairs also play an important role in the *S. maximus* spleen infected with *A. salmonicida*.

### 3.6. Construction of the circRNA-miRNA-mRNA Regulatory Network

Gene regulatory network is a complex dynamic network system, which is the overall performance of the interaction between various factors, including circRNA and miRNA. A large number of circRNA-miRNA pairs and miRNA-mRNA pairs were identified in the spleen of *S. maximus* infected with *A. salmonicida*. The circRNA-miRNA-mRNA regulatory network has been verified to play an important role in various bacterial infected fish tissues. Therefore, the triple networks of circRNA-miRNA-mRNA were constructed for exploring underlying pathogenesis of *A. salmonicida* infection by combining the data from circRNA-miRNA interactions and miRNA-mRNA interactions. After a lot of in-depth analysis, 96 circRNA-miRNA-mRNA regulatory networks (name as DE-circRNA-miRNA-mRNA) consisting of 24 DE-circRNAs, 15 overlap DE-miRNAs, and 44 target DE-mRNAs were finally obtained (Figure 4A and Appendix A). Among all the DE-circRNA-miRNA-mRNA regulatory networks, 15 of them showed significant differences in the expression of all three RNAs in the form of “up (circRNA)-down (miRNA)-up (mRNA)” or “down-up-down” after being infected with *A. salmonicida*, six were expressed in the form of “up-down-up”, and nine were expressed in the form of “down-up-down” (Figure 4B and Appendix A).

Six hours post-infection, the downregulated circRNAs novel_circ_0006815 interacted with dre-miR-138-5p and affected the expression of two target genes of dre-miR-138-5p, ras-related protein *rab-6A*, and *desmin*. The same situation also occurred at 12 h after infection. The novel_circ_0001612 and novel_circ_0000208 that interacted with upregulated dre-miR-138-5p were downregulated, and the downregulation of dre-miR-138-5p targeted keratin type I cytoskeletal 13 (*krt13*) and phosphatase-related protein type 1 (*ppr1*) was also observed. Acts as novel_324 adsorbate, the low expression of novel_circ_0002055 can hardly weaken the inhibition of novel_324 on transcription factors *mafB*, *ataxin*-2, SH3 and cysteine-rich domain-containing protein 2 (*stac2*), and cell adhesion molecule 3 (*cam3*) at 24 h post-infection. Furthermore, the novel_circ_0001830, novel_circ_0000627, novel_circ_0001881, and novel_circ_0003494 downregulated the expression of dre-miR-193a-3p, novel_187, dre-miR-210-5p, novel_685, and then the miRNA-targeted genes, including gap junction Cx32.2, glucagon-1, and CC chemokine (*ccl19*), were upregulated in the 96 h post-infection. The detailed information of circRNA-miRNA-mRNA triple networks of different time points is shown in Appendix A.

### 3.7. qRT-PCR Verification of Selected circRNAs, miRNAs, and mRNAs

qRT-PCR was used to validate the expression profiles of circRNAs, miRNAs, and mRNAs obtained by Illumina sequencing. The expression profiles of four detected DE-circRNAs (novel_circ_0000498, novel_circ_0004184, novel_circ_0002683, and novel_circ_0003361), four detected DE-miRNAs (dre-miR-233, dre-miR-200a-3p, dre-miR-200b-3p, and dre-miR-122), and eight detected DE-mRNAs (gene14346, gene658, gene9492, gene1381, gene16286, gene16422, gene19654, and gene9752) in *A. salmonicida*-infected *S. maximus* spleen were randomly selected and measured (Figure 5, Figure 6 and Figure 7).

As shown in Figure 5, most of the qRT-PCR results of four selected circRNAs were consistent with those of Illumina sequencing (correlation coefficient rs ≥ 0.9, except for novel_circ_0004184 = 0.3). As shown in Figure 6, there was similarity between the quantitative assay and high-throughput sequencing analysis of the four miRNAs in terms of fold change and significance of differential expression (correlation coefficient rs ≥ 0.6). Although there were few differences in fold change of expression, the variation trend was identical. As shown in Figure 7, all eight randomly selected DE-mRNAs showed a similar expression pattern between qRT-PCR and Illumina sequencing (correlation coefficient rs ≥ 0.66), although there were slight differences in the fold change. The results confirmed the reliability and accuracy of the high-throughput sequencing data.

For example, both quantitative assay and high-throughput sequencing analysis showed that three immune-related genes (gene16286 (CD83), gene16422 (tumor necrosis factor α -induced protein 2), and gene9752 (CC chemokine)) were upregulated in *S. maximus* spleen when infected with *A. salmonicida* (Figure 7). Novel_circ_0004184 showed downregulated expression and the other three detected circRNAs showed upregulated expression in all time points post-infection in response to *A. salmonicida* infection (Figure 5). The similar expression trends of miRNAs were also found by quantitative assay and high-throughput sequencing analysis (Figure 6). The results showed that the expression trends of most genes in qRT-PCR were in agreement with the Illumina sequencing data.

## 4. Discussion

The spleen is the primary immune tissue of teleost except kidney, thymus, and mucosa-associated lymphoid tissues. The spleen not only contains immune T cells and immune B cells that can destroy or neutralize antigens to prevent further manifestations during the disease process, but also plays an important role in antigen presentation and adaptive immune response [46,47]. Therefore, expression profiling of the spleen of *S. maximus* infected with *A. salmonicida* will be helpful to understand the mechanism of disease resistance and disease defense of *S. maximus*.

The analysis of the whole transcriptome results showed that we have obtained high quality and credible expression profile data of circRNA, miRNA, and mRNA. qRT-PCR results of randomly selected circRNAs, miRNAs, and mRNAs also confirmed that.

Non-coding RNAs are composed of a variety of RNAs that do not encode proteins and have long been known as “junk DNA”. However, many studies have shown that ncRNAs are functional RNA molecules that regulate gene expression at the transcriptional and post-transcriptional levels [48]. According to their length, ncRNAs can be divided into two categories according to the length: >200 nt and <200 nt. The ncRNAs of >200 nt are mainly long non-coding RNA, and those of <200 nt include microRNA, short interfering RNA (siRNA), and piwi-interacting RNA (piRNA) [49]. Different from the linear ncRNAs mentioned above, circRNA is a covalently closed loop of ncRNAs with large length variations [2]. These ncRNAs perform their function in various physiological and pathological conditions, including embryonic development, tissue differentiation, tissue homeostasis, disease development, and disease suppression [15,50,51,52].

These ncRNAs participate in life processes by forming regulatory networks, such as circRNA-miRNA-mRNA regulatory networks. MiR-195a bound to circRNA_28313 and CSF1 to form circRNA-miRNA-mRNA network. As a result, circRNA_28313 released the inhibitory effect of mir-195a on CSF1 and regulated osteoclast differentiation of BMM (bone marrow monocyte/macrophage) cells [53]. CircRNA_09505 could promote AKT1 expression in macrophages via IκBα/NF-κB signaling pathway by acting as a miR-6089 sponge, thereby affecting the inflammatory response involved in macrophages [54]. circRNAs showed significant regulation in Type 1 Diabetes Mellitus. Hsa_circ_0060450 could release the target gene (SHP2 and PTPN11) of miR-199a-5p and further suppressed the JAK-STAT signaling pathway triggered by type I interferon (IFN-I) to inhibit macrophage-mediated inflammation [55]. In a word, circRNA-miRNA-mRNA regulatory networks existed and played a pivotal role in a variety of physiological and pathological processes. However, there were relatively few studies on the regulatory role of circRNA and miRNA in teleost fish after pathogen infection. Therefore, we systematically analyzed the circRNA, miRNA, and mRNA expression profiles of *S. maximus* spleen infected with *A. salmonicida*.

A total of 119 DE-circRNAs, 140 DE-miRNAs, and 510 DE-mRNAs were identified at four infection time points in *S. maximus* exposed to *A. salmonicida* infection. It has been found that these differential RNAs constituted multiple circRNA-miRNA-mRNA regulatory networks.

Similarly, 87 DE-circRNAs were identified by expression profile analysis in the intestine of *Sebastes schlegelii* after pathogen infection. The circRNA-miRNA-mRNA regulation network was found involved in the regulation of the NF-kB signaling pathway and the expression of *interleukin 1 receptor* [32]. Novel_circ_0004195, serving as ceRNA of novel_530, could activate the downstream NF-kB signaling pathway. It was demonstrated that circRNAs play an important antibacterial role in the intestinal tissues of *Sebastes schlegelii* challenged with *E. tarda* by forming the circRNA-miRNA-mRNA network.

In our study, circRNA was also found to regulate the expression of immune genes acting as miRNA sponges. Just as found in the constructed circRNA-miRNA and miRNA-mRNA regulatory networks, immune-related genes (*gap junction Cx*32.2, *cell adhesion molecule* 3, and *CC chemokine*) were also found in DE-circRNA-miRNA-mRNA regulatory networks (Figure 4B). These results indicated that circRNAs and miRNAs might play an important role in pathogenic bacteria-infected *S. maximus* in the form of circRNA-miRNA-mRNA network.

*Connexin*32 (*Cx*32) is a member of the gap junction protein family, which serves a useful function in the mammalian immune system. There is research suggesting that the expression of LPS-induced pro-inflammatory cytokines *IL-8* and *TNF-α* was significantly reduced by *Cx*32 in Japanese flounder (*Paralichthys olivaceus*), indicating the important immunological functions of *Cx*32 [56]. The expression of dre-miR-193a-3p was decreased in the spleen of *S. maximus* infected with *A. salmonicida* for 96 h, and novel_circ_0001830 and novel_circ_0000627, which interacted with dre-miR-193a-3p, were highly expressed. Meanwhile, the expression of *connexin*32, which was predicted and targeted by dre-miR-193p, was also increased. Therefore, we speculated that novel_circ_0001830 and novel_circ_0000627 could indirectly regulate *connexin*32 to resist the invasion of pathogenic bacteria.

Chemokines are a family of cytokines that regulate the migration and immune response of leukocytes [57,58]. Chemokine (CC motif) ligand 19 (*CCL*19) is a cytokine that belongs to the CC chemokine subfamily. It is a key regulator of T cell activation, immune tolerance, and inflammatory response [59]. *CCL*19 could be targeted and inhibited by miR-325-3p to attenuate renal inflammation in diabetic nephropathy mice [60]. The expression level of *ccl*19 in the spleen of *S. maximus* infected with *A. salmonicida* was increased. *ccl*19 was reported to promote inflammation by activating Mo/Mφ and lymphocytes in pathogen-infected teleost [61]. Recombinant ccl19 could stimulate the expression of cytokines in the mucosal system and the proliferation of leukocytes in head kidney [62]. In *S. maximus*, *ccl*19 was also reported to induce leukocyte trafficking and promote anti-viral and anti-bacterial defense [63].

In our study, the expression of *ccl*19 was up-regulated in the spleen of *S. maximus* infected with *A. salmonicida* for 96 h, and the novel_685 that targeted *ccl*19 was down-regulated, while the predicted sponge of novel_685 and novel_circ_0001881 was up-regulated at the same time. *ccl*19, novel_685, and novel_circ_0001881 formed a circRNA-miRNA-mRNA regulatory network. This indicated that circRNAs and miRNAs could participate in the anti-pathogen immune response in *S. maximus* infected with *A. salmonicida* through the formation of circRNA-miRNA-mRNA regulatory network.

## 5. Conclusions

In conclusion, we analyzed the expression profile of *Scophthalmus maximus* spleen infected with *A. salmonicida* and constructed the circRNA-miRNA-mRNA regulatory networks. The establishment of circRNA-miRNA and miRNA-mRNA regulatory networks provided evidence that circRNA could participate in the immune regulation of fish through the circRNA-miRNA-mRNA regulatory network. It also provided a new insight into the immune response pathway of fish after pathogen infection. These immune-related circRNA-miRNA-mRNA regulatory networks still need to be further validated in order to understand the immune mechanism of *S. maximus* against *A. salmonicida* from a broader perspective.

## Figures and Tables

**Figure 1 biology-10-00626-f001:**
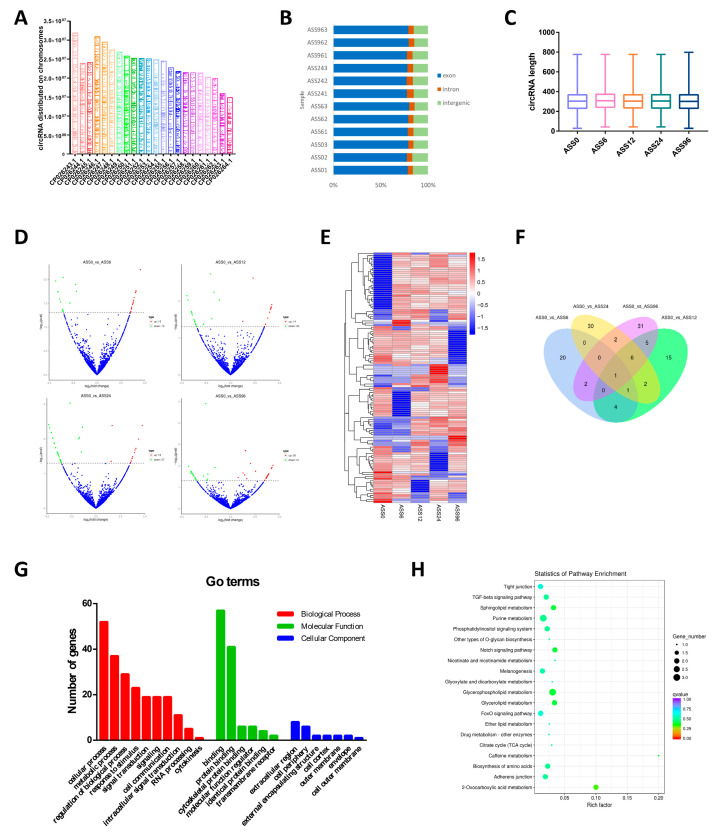
Analysis of circRNA sequencing data. (**A**) The distribution of circRNAs on turbot different chromosomes; the box represents the length of the chromosomes. (**B**) The number of circRNAs arising from different genomic locus (exon, intronic, and intergenic). (**C**) Size distribution of unique circRNA sequences from ASS0, ASS6, ASS12, ASS24, and ASS96 groups. (**D**) Volcano Plots of DE-circRNAs among ASS0 and infected groups. Red blocks represent up-regulated circRNAs and green blocks represent down-regulated circRNAs. (**E**) Heatmap was used to assess the expression of DE-circRNAs. Red and blue denote high and low expression, respectively. Each DE-circRNA is represented by a single row of colored boxes, and each group is represented by a single column. (**F**) Pairwise comparisons of differentially expressed circRNAs between two groups. The Venn diagrams display the distribution of the 119 unique DE-circRNAs between ASS6, ASS12, ASS24, ASS96, and ASS0, respectively. (**G**) Go term analysis of DE-circRNAs. (**H**) KEGG analysis of DE-circRNAs.

**Figure 2 biology-10-00626-f002:**
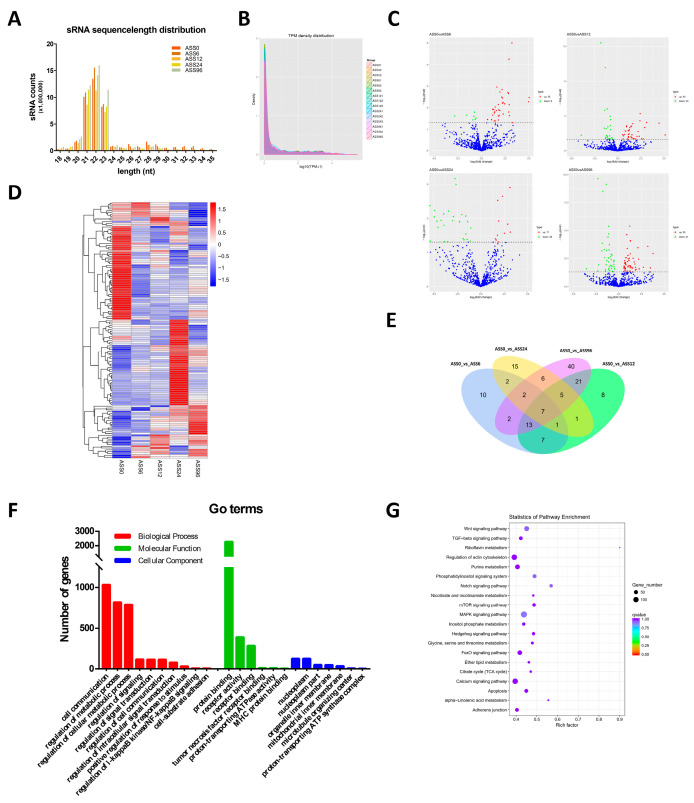
Analysis of miRNA sequencing data. (**A**) The length of miRNAs from ASS0, ASS6, ASS12, ASS24, and ASS96 groups. (**B**) TPM density distribution of sRNAs from 15 samples. (**C**) Volcano Plots of DE-miRNAs among ASS0 and infected groups. Red blocks represent up-regulated miRNAs and green blocks represent down-regulated miRNAs. (**D**) Heatmap was used to assess the expression of DE-miRNAs. Red and blue denote high and low expression, respectively. Each DE-miRNA is represented by a single row of colored boxes, and each group is represented by a single column. (**E**) Pairwise comparisons of differentially expressed miRNAs between two groups. The Venn diagrams display the distribution of the 140 unique DE-miRNAs between ASS6, ASS12, ASS24, ASS96, and ASS0, respectively. (**F**) Go term analysis of DE-miRNAs. (**G**) KEGG analysis of DE-miRNAs.

**Figure 3 biology-10-00626-f003:**
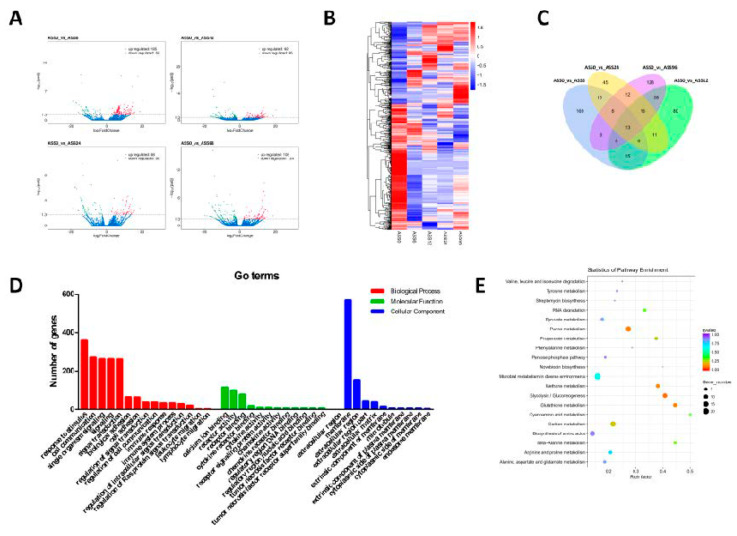
Analysis of mRNA sequencing data. (**A**) Volcano Plots of DE-mRNAs among ASS0 and infected groups. Red blocks represent up-regulated mRNAs and green blocks represent down-regulated mRNAs. (**B**) Heatmap was used to assess the expression of DE-mRNAs. Red and blue denote high and low expression, respectively. Each DE-mRNA is represented by a single row of colored boxes, and each group is represented by a single column. (**C**) Pairwise comparisons of differentially expressed mRNAs between two groups. The Venn diagrams display the distribution of the 530 unique DE-mRNAs between ASS6, ASS12, ASS24, ASS96, and ASS0, respectively. (**D**) Go term analysis of DE-mRNAs. (**E**) KEGG analysis of DE-mRNAs.

**Figure 4 biology-10-00626-f004:**
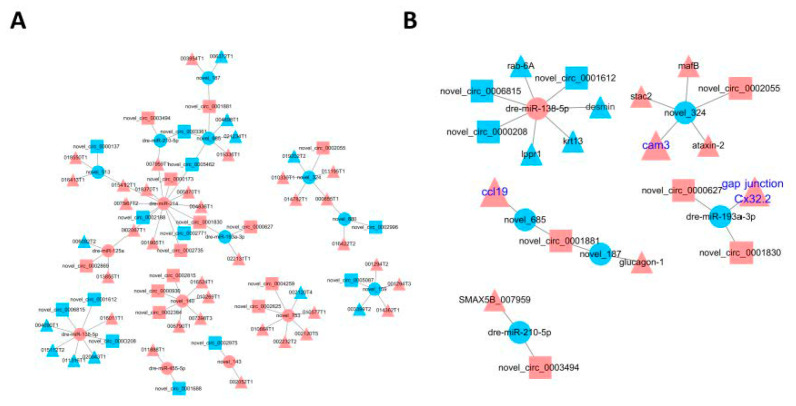
CircRNA-miRNA-mRNA regulatory network. (**A**) The DE-circRNA-miRNA-mRNA regulatory networks found in ASS6, ASS12, ASS24, and ASS96 groups compared with ASS0. (**B**) The DE-circRNA-miRNA-mRNA regulatory network in the form of “up-down-up” (circRNA-miRNA-mRNA) or “down-up-down” after infection. CircRNAs, miRNAs, and mRNA are indicated by square, circle, and triangle, respectively; red represents up regulation and blue represents down regulation.

**Figure 5 biology-10-00626-f005:**
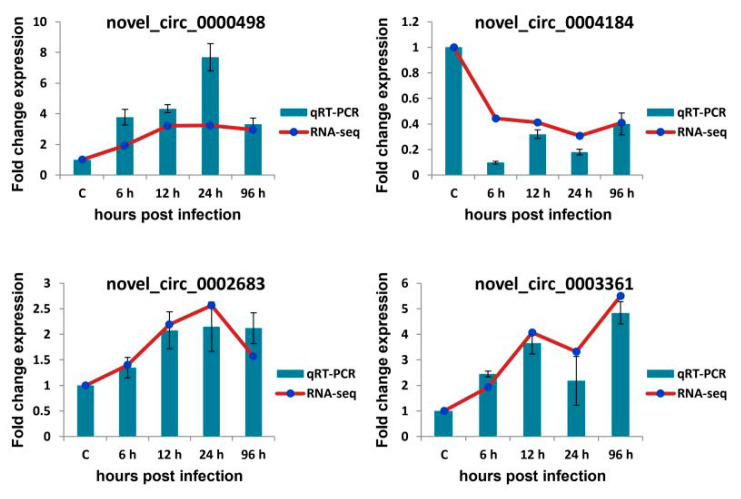
Verification of circRNA expression level by qRT-PCR. Four circRNAs were randomly selected for qRT-qPCR analysis and compared with data obtained using RNA-sequencing (RNA-seq). The relative expression levels of circRNAs in ASS6, ASS12, ASS24, and ASS96 groups were calculated as the ratio of the gene expression level (qRT-PCR) or normalized transcripts per million (TPM) (RNA-seq) relative to ASS0 group. The qRT-PCR data are given as means ± standard deviation of three replicates.

**Figure 6 biology-10-00626-f006:**
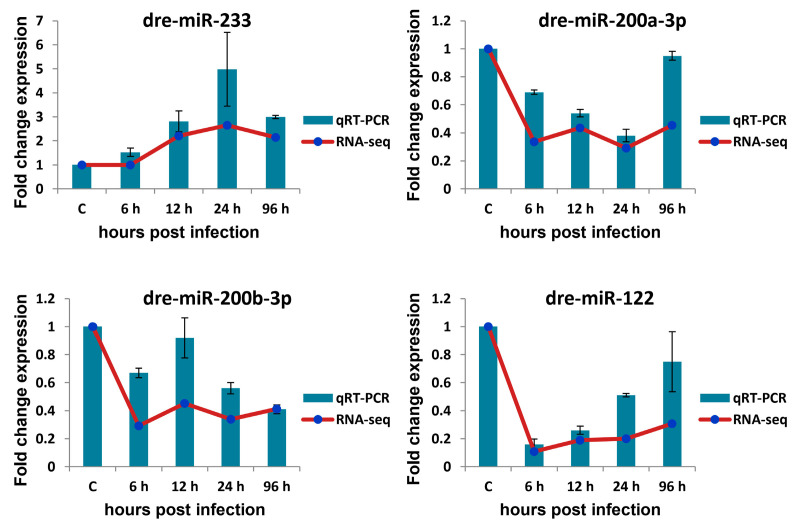
Verification of miRNA expression level by qRT-PCR. Four miRNAs were randomly selected for qRT-qPCR analysis and compared with data obtained using RNA-sequencing (RNA-seq). The relative expression level of miRNAs in ASS6, ASS12, ASS24, and ASS96 groups was calculated as the ratio of the gene expression level (qRT-PCR) or normalized transcripts per million (TPM) (RNA-seq) relative to the ASS0 group. The qRT-PCR data are given as means ± standard deviation of three replicates.

**Figure 7 biology-10-00626-f007:**
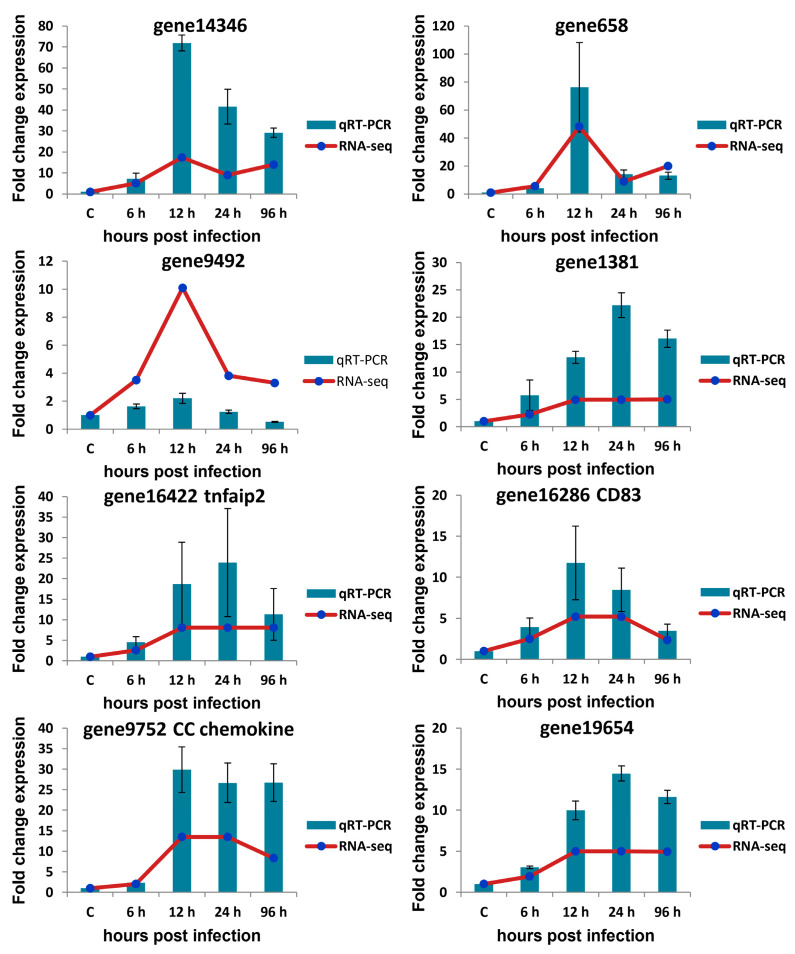
Verification of mRNA expression level by qRT-PCR. Eight mRNAs were randomly selected for qRT-qPCR analysis and compared with data obtained using RNA-sequencing (RNA-seq). The relative expression level of mRNAs in ASS6, ASS12, ASS24, and ASS96 groups was calculated as the ratio of the gene expression level (qRT-PCR) or normalized transcripts per million (TPM) (RNA-seq) relative to the ASS0 group. The qRT-PCR data are given as means ± standard deviation of three replicates. tnfaip2: tumor necrosis factor alpha-induced protein 2.

**Table 1 biology-10-00626-t001:** Information list of circRNA sequencing data.

Sample_Name	Raw_Reads	Clean_Reads	Clean_Bases (Gb)	Error Rate (%)	Q20 (%)	Q30 (%)	GC_Content (%)	Mapped to Genome (%)
ASS01	89,427,870	87,674,318	13.15	0.03	97.22	92.58	54.11	89.81
ASS02	88,835,296	86,925,034	13.04	0.03	97.32	92.84	51.33	90.62
ASS03	91,281,976	87,308,908	13.10	0.03	97.01	92.23	53.22	86.94
ASS61	84,909,274	78,465,246	11.77	0.03	96.81	92.12	55.28	90.44
ASS62	95,743,386	93,531,870	14.03	0.03	97.37	92.93	52.17	87.85
ASS63	89,110,602	87,785,728	13.17	0.03	97.51	93.21	52.94	90.53
ASS121	97,847,968	96,072,446	14.41	0.03	97.42	92.93	53.21	85.35
ASS122	107,925,458	106,515,648	15.98	0.03	97.26	92.60	53.04	87.26
ASS123	88,551,056	86,735,528	13.01	0.03	97.24	92.61	52.54	90.15
ASS241	108,968,470	107,506,050	16.13	0.03	97.26	92.67	55.74	80.36
ASS242	93,831,136	92,152,780	13.82	0.03	97.41	92.96	51.69	85.96
ASS243	86,686,374	84,702,798	12.71	0.03	97.29	92.76	53.08	88.55
ASS961	100,039,326	98,296,586	14.74	0.03	97.68	93.61	54.47	88.51
ASS962	87,993,170	83,798,128	12.57	0.03	96.98	92.24	53.90	84.57
ASS963	104,529,398	102,835,526	15.43	0.03	97.62	93.43	53.81	84.48

**Table 2 biology-10-00626-t002:** Information list of miRNA sequencing data.

Sample	Total_Reads	Clean Reads	Error Rate (%)	GC Content (%)	Mapped sRNA (%)	Uniq Reads
ASS01	16,992,257	16,776,891	0.01	48.91	94.56	225,970
ASS02	13,585,729	13,445,193	0.01	49.72	91.22	256,756
ASS03	14,090,926	13,924,028	0.01	49.17	93.82	233,122
ASS61	16,759,631	16,480,923	0.01	48.43	94.80	292,697
ASS62	14,970,408	14,487,862	0.01	49.27	90.79	297,090
ASS63	17,691,027	17,287,838	0.01	48.97	93.65	380,538
ASS121	15,944,207	15,759,031	0.01	48.90	95.73	280,956
ASS122	14,993,055	14,650,290	0.01	49.43	92.89	331,440
ASS123	14,495,408	10,351,418	0.01	51.08	89.65	172,990
ASS241	19,726,929	19,222,454	0.01	48.95	95.82	276,881
ASS242	13,696,253	12,062,266	0.01	54.28	66.84	301,900
ASS243	15,432,974	15,212,095	0.01	48.60	95.28	305,646
ASS961	16,295,413	16,051,491	0.01	48.70	95.38	326,589
ASS962	18,209,628	17,873,027	0.01	48.35	94.96	376,083
ASS963	17,714,908	17,364,745	0.01	49.44	93.82	319,525

**Table 3 biology-10-00626-t003:** Information of identified miRNAs.

Types	Known miRNA				New miRNA				
	Mapped Mature	Mapped Hairpin	Mapped Uniq sRNA	Mapped Total sRNA	Mapped Mature	Mapped Star	Mapped Hairpin	Mapped Uniq sRNA	Mapped Total sRNA
ASS01	214	258	2539	10,109,019	354	159	422	1721	849,828
ASS02	206	255	2394	7,201,071	330	143	400	1504	486,725
ASS03	212	257	2559	8,142,478	355	173	432	1714	577,499
ASS121	218	263	2643	10,991,726	389	180	464	1910	721,131
ASS122	210	257	2468	8,730,453	344	167	404	1690	641,866
ASS123	202	237	1834	3,157,360	234	112	305	1036	234,731
ASS241	216	245	2906	12,630,412	393	185	479	2055	974,386
ASS242	179	223	1272	1,934,747	207	117	282	810	154,078
ASS243	211	258	2573	10,726,049	397	174	474	1935	776,575
ASS61	208	271	2446	10,849,391	396	168	462	1829	779,494
ASS62	203	249	2401	7,794,581	318	151	392	1613	356,696
ASS63	211	256	2559	11,483,832	399	167	470	1867	693,347
ASS961	211	241	2690	10,345,787	395	186	464	2005	880,402
ASS962	212	259	2602	12,439,717	415	198	493	2073	983,840
ASS963	214	256	2720	10,828,035	394	178	472	1941	845,700
Total	252	286	65,488	245,032,276	668	442	726	45,784	16,824,392

## Data Availability

The data presented in this study are available in this article or Appendix A.

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
