# Peer review of "Revealing New Landscape of Turbot (Scophthalmus maximus) Spleen Infected with Aeromonas salmonicida through Immune Related circRNA-miRNA-mRNA Axis"

_biology, 2021, doi:10.3390/biology10070626_

Round 1
Reviewer 1 Report
In this manuscript, Xue et al. challenged turbot with Aeromonas salmonicida to evalute the immune response in spleen tissue by high-throughput sequencing. The results show a regulation of a network compromising circRNA-miRNA-mRNA of several genes, including genes with immune functions, thay may potentially be involve in such regulation.
This a well-done RNA-seq study that contributes to increase the knowledge in its respective field of research. The scientific approach is rigorous and solid. Conclusions are correct according to the data that authors have shown in this paper. However, I have some concerns after reviewing the manuscript:
Major
- In general, the size of the figures across the manuscript is very small. Sometimes, it is not even possible to read the Figure legends. All figures should be generated again with a higher quality: more DPI on each figure and a proper size.
- Figures 5, 6 and 7 lack a statistical analysis that correlates the tendencies observed by RNA-Seq and RT-qPCR between the different samples. Although the trend between both techniques seems clear, there must be a statistical test behind to support such claim. I suggest to re-analyze these data with a Spearman's nonparametric correlation.
Minor
- Why were the spleen tissues pooled? Immune responses tend to vary greatly between individuals in some cases. Why did the authors discard the possibility of analyzing the immune response on individual fish? The authors should provide a statement clarifying why they decided to pooled the samples.
- Line 69: 'some of these research has...' should be considered to be changed to 'some of these studies have...'. It sounds grammatically incorrect to me.
- Material and methods, Turbot treatment and sampling collection: how turbot were euthanized is not indicated in this section.
- Table 1 and 2, the 'error rate' parameter is not described. In fact, it is represented as a percentage in Table 2 while in Table 1 in shown as a decimal number (without percentage). The authors must clarify its meaning in the Figure caption.
Reviewer 2 Report
The manuscript by Xung et al. investigated the expression of circular RNAs (circRNAs), microRNAs (mirRNAs), and MRNAs in the spleen of Scophthalmus maximus, experimentally infected with Aeromonas salmonicida. The circRNAs-mirRNAs-MRNAs network was constructed and exploited to investigate the role of non-coding RNA in the immune system of S. maximus. CircRNAs are known to participate in several biological processes, especially in cell regulation, signaling pathway regulation, as well tissue development. However, recent research shows that circRNAs are also involved in the innate immune response. What highlights the importance of the study.
The manuscript is of quality and relevant to aquaculture. The introduction is well-written, comprehensive, and conceptualizes the reader about the importance of circRNAs for teleosts. The material and methods are adequate, the results were well presented, and the discussion and conclusion appropriate. However, I suggest that minor revisions be made before recommending it for publication in Biology.
Line 37: I suggest that you avoid words that are already present in the manuscript title.
*I suggest that the dpi of figures 1, 2, 3 and 4 be adjusted to improve sharpness.
